# Joint profiling of DNA and proteins in single cells to dissect genotype-phenotype associations in leukemia

Benjamin Demaree [1,2,10], Cyrille L. Delley [1,10], Harish N. Vasudevan[1,3], Cheryl A. C. Peretz[4,5,6], David Ruff[7], Catherine C. Smith[4,8] & Adam R. Abate [1,2,9✉]

Studies of acute myeloid leukemia rely on DNA sequencing and immunophenotyping by flow cytometry as primary tools for disease characterization. However, leukemia tumor heterogeneity complicates integration of DNA variants and immunophenotypes from separate measurements. Here we introduce DAb-seq, a technology for simultaneous capture of DNA genotype and cell surface phenotype from single cells at high throughput, enabling direct profiling of proteogenomic states in tens of thousands of cells. To demonstrate the approach, we analyze the disease of three patients with leukemia over multiple treatment timepoints and disease recurrences. We observe complex genotype-phenotype dynamics that illustrate the subtlety of the disease process and the degree of incongruity between blast cell genotype and phenotype in different clinical scenarios. Our results highlight the importance of combined single-cell DNA and protein measurements to fully characterize the heterogeneity of leukemia.

[1] Department of Bioengineering and Therapeutic Sciences, University of California, San Francisco, San Francisco, CA, USA. [2] UC Berkeley-UCSF Graduate Program in Bioengineering, University of California, San Francisco, San Francisco, CA, USA. [3] Department of Radiation Oncology, University of California, San Francisco, San Francisco, CA, USA. [4] Division of Hematology/Oncology, Department of Medicine, University of California, San Francisco, San Francisco, CA, USA. [5] Children's Hospital and Research Center Oakland, Oakland, CA, USA. [6] Department of Pediatrics, University of California, San Francisco, San Francisco, CA, USA. [7] Mission Bio, Inc., South San Francisco, CA, USA. [8] Helen Diller Family Comprehensive Cancer Center, University of California, San Francisco, San Francisco, CA, USA. [9] Chan Zuckerberg Biohub, San Francisco, CA, USA. [10] These authors contributed equally: Benjamin Demaree, Cyrille L. Delley. ✉email: adam@abatelab.org

Cellular heterogeneity is an intrinsic aspect of cancer that drives disease progression and relapse. Because cancer cells are heterogeneous in genotype and phenotype, it is difficult to directly link genotypes to immunophenotypes beyond circumstantial evidence from epidemiologic studies[1,2]. In acute myeloid leukemia (AML), an aggressive hematologic malignancy, this heterogeneity manifests as polyclonal cancer cells with distinctive genotypes but diverse immunophenotypes[3,4]. While leukemic blasts often exhibit immunophenotypes distinct from normal cells, with some surface markers even serving as therapeutic targets[5], immunotypes are only weakly prognostic. Genotypes, in contrast, are more informative on the disease course, which suggests a weak correspondence between these domains[1,2].

Single-cell analysis provides a powerful tool for characterizing this complexity, and thereby advancing our understanding of different cancers. The value of multiomic single-cell analysis is its ability to correlate co-occurrence of different features in individual cells, with high-throughput technologies permitting analysis of thousands of cells to generate rich and intricate feature maps. For example, single-cell genotyping of AML-relevant loci has revealed co-occurrence of mutations and mapping of the clonal relationships between blasts[6–9]. These studies, however, have yet to map DNA variants and surface phenotypes in the same cells, precluding direct linkage of phenotypes to the genetic mutations that drive them.

To obtain simultaneous genotype and immunophenotype information, single cells can be sorted based on multi-parametric antibody analysis, and sequenced. While limited in throughput, these studies have uncovered important insights into the genetics of AML, identifying relevant aberrations, such as single-nucleotide polymorphisms and gene fusions[10]. Single-cell RNA sequencing (scRNA-seq) has emerged as a potentially valuable approach for genotype–phenotype linkage because it is cost effective and scalable[7,11–13]. The mRNA sequences provide genotype information[13,14], while the abundance of these sequences yields phenotypic information[15–19]. Moreover, modern approaches are high throughput, allowing characterization of thousands of cells. Nevertheless, genotyping from mRNA remains a challenging and error-prone procedure that, even in the best case, provides incomplete information. For example, stochastic gene expression, biological biases[20], and limited coverage of essential genes combine to make assigning a genotype more difficult than can be achieved by direct analysis of DNA. Moreover, since RNA methods analyze only the expressed portion of the genome, mutations in intronic and other non-transcribed elements, like transcription factor binding sites, are omitted[21,22]. Thus, while several technologies have highlighted the importance of high-throughput single-cell genotype–phenotype measurements, none provide the scalability and precision for comprehensive and accurate mapping of these biomarkers.

Here, we describe DAb-seq, a tool for joint profiling of DNA and surface proteins in single cells at high throughput. While existing methods attempt to obtain this information from the transcriptome alone, our approach directly sequences DNA for genotype and surface proteins for phenotype—both gold standards in blood cancer studies. DAb-seq is thus distinct from RNA and antibody sequencing methods, which capture phenotypic features of cells, but do not yield genotypes directly from DNA. To illustrate the power of DAb-seq, we investigate the immunophenotypic and genotypic diversity underpinning AML in three patients at multiple timepoints. We leverage the method's throughput to analyze 49 DNA targets and 23 hematopoietic markers in a total of 54,717 cells. This analysis allows tracking of proteogenomic dynamics over multiple treatments and recurrences. We identify cases of genotype–phenotype decoupling, observing immunophenotypic heterogeneity among cells with a shared pathogenic mutation and genotypically diverse cells with a convergent malignant immunophenotype. These findings indicate variability of blast fate upon treatment in AML. Furthermore, our results underscore that independent phenotype or genotype measurements do not adequately capture proteogenomic heterogeneity. More broadly, our work demonstrates how single-cell technologies can elucidate the complex interplay between DNA mutations and their effects on protein expression in cancer.

## Results

To achieve combined DNA sequencing and antibody profiling in single cells, we developed a workflow leveraging a commercial microfluidic platform. The Mission Bio Tapestri instrument is designed for targeted sequencing of thousands of single cells, and has been applied to DNA genotyping and lineage mapping in cancers[8]. We modified its protocol to support Abseq, a separate method we developed[23] that allows characterization of single-cell surface proteins by sequencing. As in Abseq, DAb-seq begins with immunostaining of a cell suspension using a mixture of antibody–oligo conjugates (Fig. 1a). Each antibody is associated with a known oligo tag; thus, when cells are stained with a pool of tagged antibodies, each cell is bound with a combination of antibodies and their tags based on surface protein profile.

The stained cells are processed through a series of microfluidic devices on the Tapestri instrument to amplify and barcode genomic targets and surface-bound antibody tags. The workflow follows a two-step protocol to lyse cells and digest chromatin, making the genome accessible to amplification; the droplets are then subjected to a multiplex PCR to simultaneously amplify the genomic targets and capture antibody tags, labeling both with a barcode fragment relating sequences from the same cell (Fig. 1b). For genotype, we target recurrently mutated genomic DNA loci in AML with primers containing a unique cell barcode against 49

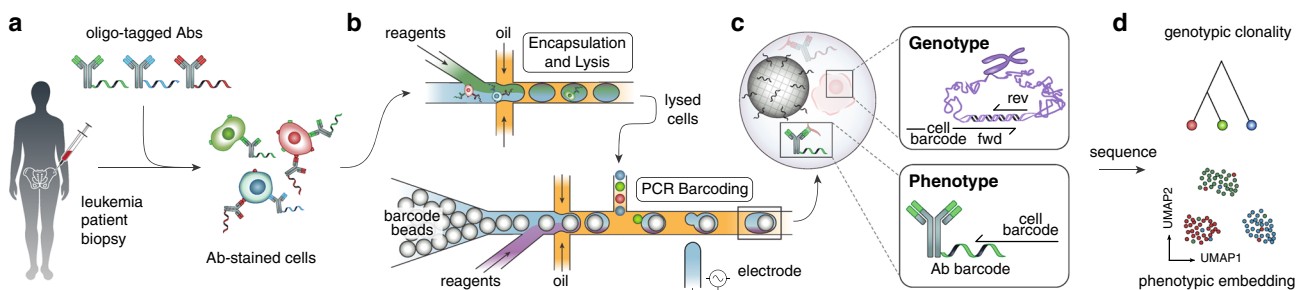

**Fig. 1 The DAb-seq workflow. a** Bone marrow aspirates of patients with AML contain healthy and malignant cells that exhibit diverse genotypes and immunophenotypes. These cells are stained with antibodies labeled with DNA tags. **b** Stained cells are paired and encapsulated with a barcode bead on a Mission Bio Tapestri instrument. **c** In each droplet, a PCR labels antibody tags and genomic DNA targets simultaneously with a unique cell index. **d** Sequencing the barcoded amplicons and antibody tags yields coupled single-cell immunophenotype and genotype data for thousands of cells.

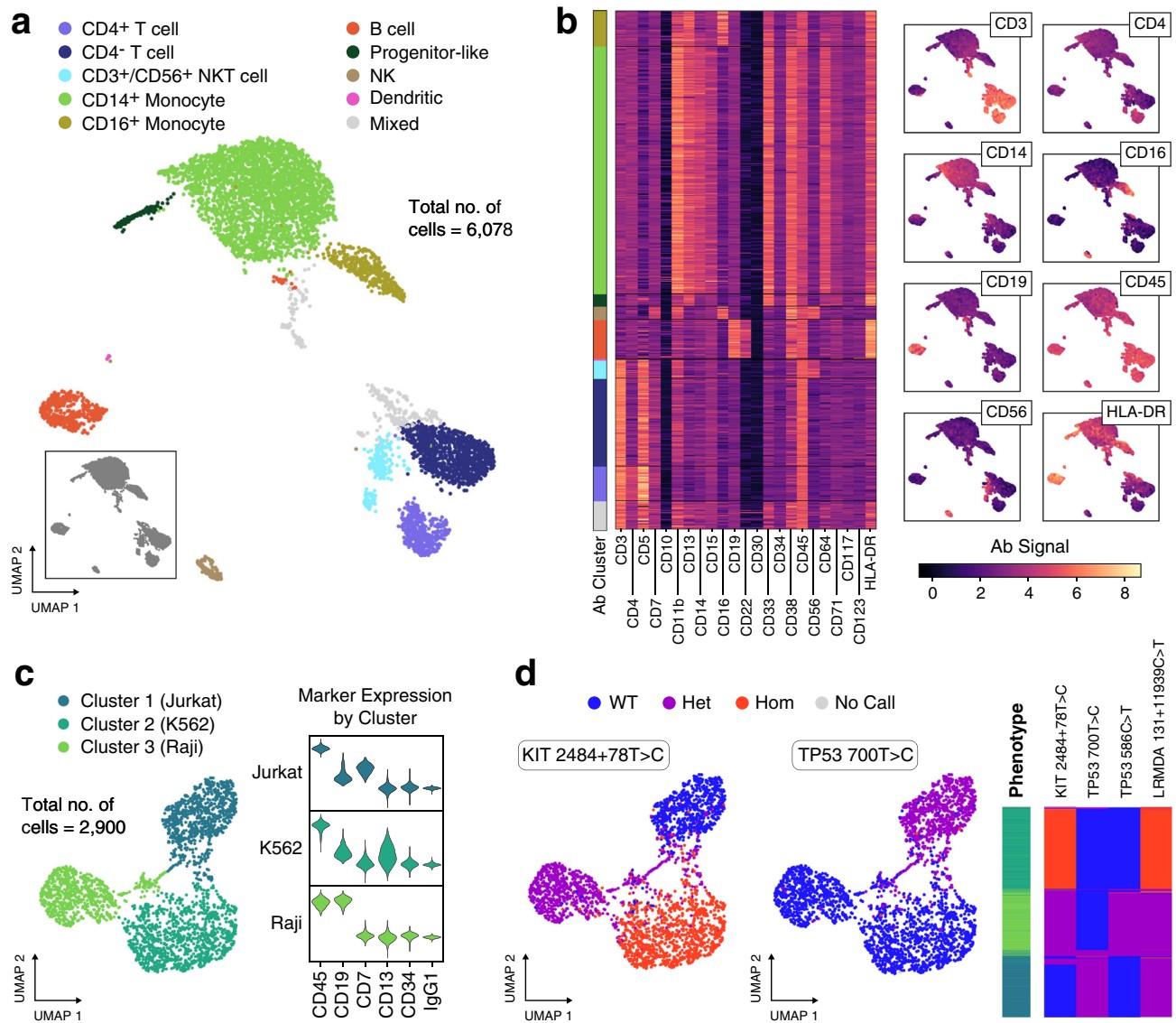

**Fig. 2 DAb-seq enables simultaneous discrimination of single cells by their immunophenotype and genotype. a** DAb-seq workflow performed on PBMCs from a healthy donor using a panel of 23 antibodies. Leiden clustering and two-dimensional UMAP embedding of the antibody tag data reveals expected blood cell populations. Cell clusters are manually annotated based on detected marker expression. **b** Heatmap of the corrected log-transformed antibody counts for each cell and antibody. Cells are ordered based on Leiden clusters. Overlay of corrected log-transformed antibody counts with the UMAP embedding highlights compartment-specific expression. **c** Correspondence of antibody signal with genomic polymorphisms in DAb-seq experiments tested on a mixture of three cell lines and a panel of six antibodies. Cells cluster by antibody signal as shown in the UMAP embedding. **d** Detected single-nucleotide polymorphisms in these cells map to the phenotypic cell clusters as shown in the UMAP embedding and a heatmap, where rows correspond to single cells. The first column of the heatmap indicates assigned phenotype cluster, and the remaining columns indicate the genotyping call at the labeled loci. Source data are provided as a Source data file.

amplicons spanning 19 genes (Supplementary Table 1). The primers and PCR conditions are tuned to enable uniform and quantitative amplification of all genomic DNA targets (Supplementary Fig. 1). These primers also capture antibody tags from a 23-plex immunophenotyping panel based on those used in clinical minimal residual disease studies[24,25] (Fig. 1c and Supplementary Table 2). Sequencing yields a multiomic data set where each cell is represented by a genotype and immunophenotype vector. This data can be visualized as a low-dimensional embedding and subjected to unsupervised clustering to identify cell populations with a similar immunophenotype (Fig. 1d).

Peripheral blood mononuclear cells (PBMCs) comprise a diverse cell population and provide a sample with which to assess the effectiveness of DAb-seq for mapping hematopoietic

immunophenotypes. When applied to PBMCs from a healthy donor, we obtain expected cell subsets across blood compartments, identifying both rare and abundant cells in peripheral blood (Fig. 2a, b and Supplementary Fig. 2), with gated populations matching those obtained by flow cytometry performed in parallel (Supplementary Fig. 3). Because healthy PBMCs should harbor no pathogenic mutations, they serve as a negative control for DNA genotyping purposes, and we find no pathogenic variants across the targets analyzed (Supplementary Table 3). To further validate the genotyping capability of DAb-seq, we use a mixture of three cell lines derived from distinct hematopoietic lineages (Jurkat, Raji, K562) with documented mutations in the targeted genomic regions covered by our single-cell DNA sequencing panel[26]. For all genetic variants, we assign genotype

calls to each individual cell: homozygous wild type, heterozygous alternate, or homozygous alternate. We observe the expected concordance between single-cell genotypes and phenotypes, as cells of the same genotype segregate within a common immunophenotypic cluster (Fig. 2c, d and Supplementary Table 8). We use the adjusted Rand index (ARI) as a measure of concordance between phenotype clusters and genetic variants, yielding an ARI of 0.74 for the mapping between combined genotype calls and phenotype cluster. Notably, we find that DAb-seq's genotyping is sufficiently sensitive to differentiate the cells based on zygosity of a given mutation (Fig. 2d; e.g., 0.73 ARI between KIT 2484 + 78 T > C zygosity and phenotypic cluster). These results show that DAb-seq can simultaneously profile cell genotype from targeted amplification of genomic DNA and immunophenotype from barcoded antibodies.

**NPM1-mutated cells persist across therapy timepoints with a static immunophenotype**. AML therapies targeted to cell surface proteins require homogeneous expression of the target marker across all malignant cells. We therefore hypothesized that mutated cells should robustly associate with a common targeted phenotype in patients responsive to such therapy. To investigate this, we performed DAb-seq on 21,952 total cells from bone marrow aspirates of a patient with AML receiving gemtuzumab, a CD33-targeted therapy, across four treatment timepoints (Fig. 3a). This patient received multiple rounds of chemotherapy, including a stem cell transplantation, prior to the first timepoint sampled in this study (Supplementary Table 4). From single-cell DNA genotyping data, we identify a persistent frameshift mutation in the $NPM1$ gene ($NPM1^{mut}$) across relapse, salvage therapy, and progression timepoints. In addition, the $NPM1$ mutation is found to co-occur with a mutation at the $DNMT3A$ locus (Fig. 3a; ARI between $NPM1$ and $DNMT3A$ clones: 0.61). Gemtuzumab targets $CD33^{+}$ cells, which are extinguished at the remission timepoint[27]. To examine the immunophenotypic profile of the $NPM1^{mut}$ cell population, we plot single-cell CD33 and CD34 values with $NPM1$ mutation status across timepoints (Fig. 3b). The proportion of $NPM1^{mut}$ cells in the $CD34^{-}$ and $CD34^{+}$ compartments does not vary extensively across treatments, while $CD33^{+}$ myeloid cells targeted by the drug are absent at remission, consistent with the treatment response.

In all timepoints for this patient, our analysis suggests a correspondence between the blast genotype and corresponding phenotype (ARI: 0.68; uncertainty coefficient for malignant phenotype given genotype for $NPM1$ and $DNMT3A$ clones: 0.87). To further explore this relationship between genotype and phenotype, we visualize the high-dimensional single-cell immunophenotype as a Uniform Manifold Approximation and Projection[28] (UMAP) embedding of the antibody data (Fig. 3c, and Supplementary Figs. 4 and 5). Cells within single immunophenotypic clusters originate from different timepoints, highlighting the stability of normal and malignant immunophenotypes over time. When we overlay $NPM1$ genotype onto immunophenotypic UMAP space, we find a clear association between a single malignant immunophenotype composed of $CD33^{+}$ cells with $NPM1$ mutation status, with variable expression of CD34, CD38, and CD117 in this population (Fig. 3d). Indeed, this is in agreement with previous observations in flow cytometric studies where blast cells have been found to uniformly express CD33 and variably express CD34, CD38, and CD117 (ref. [29]). Among the $NPM1^{wt}$ cells, we identify classical blood cell markers, including CD3 and CD5 (lymphocyte), CD15 (monocyte), and CD56 (natural killer). Taken together, in this patient, DAb-seq confirms elimination of $CD33^{+}$ cells by gemtuzumab treatment and reveals a strong correspondence between genotype and phenotype across timepoints.

**Genotypic subclones form overlapping subsets across an immunophenotypic continuum**. To investigate whether such tight genotype–phenotype association is a universal feature of AML, we applied DAb-seq to a pediatric patient who underwent induction and consolidation chemotherapy, but ultimately relapsed (Supplementary Table 4). We identify two mutually exclusive $KRAS$ and $FLT3$-mutated clones at diagnosis and relapse ($KRAS^{mut}$ and $FLT3^{mut}$; ARI between $KRAS$ and $FLT3$ clones: 0.07). The $FLT3^{mut}$ population, although the minor subclone at diagnosis comprising just 43 of 4563 cells (0.94%) compared to 1539 cells (33.7%) for the $KRAS^{mut}$ variant, dominates at relapse (6800 of 7516 cells, 90.5%; Fig. 4a). Immunophenotypically, we also identify a third subset comprising $KRAS^{WT}/FLT3^{WT}$ cells expressing a blast-like $CD33^{+}CD38^{+}$ immunophenotype with no identifiable DNA mutations in the targeted loci. When we group cells from all timepoints by genotype, pathogenic blasts display variable immunophenotypes, with no clear mapping between the two (Fig. 4b; uncertainty coefficient for $FLT3$ or $KRAS$ genotype given phenotype cluster: 0.16; 0.41 for malignant phenotype given $FLT3$ and $KRAS$ genotype).

In the absence of an obvious genotype–phenotype mapping for this sample, we sought to investigate the underlying relationship between these domains. Using UMAP, we project the antibody data into two dimensions, coloring the points according to genotype (Fig. 4c, and Supplementary Figs. 6 and 7). We observe a single immunophenotypic compartment with incomplete separation between genotypes. To estimate antibody profile expression within the blast compartment continuum, we identify the dominant gradients in the phenotypic space, and order all cells along the gradients. We then calculate the local average antibody and genotypic composition for neighboring cells (Fig. 4c, d and Supplementary Fig. 8; "Methods" section). As expected, some markers are anticorrelated (CD11b, CD33, and CD56) or correlated (CD15) with the principal immunophenotypic gradient (Pearson correlation: −0.78, −0.70, −0.60, and 0.83, respectively; all $p < 10^{-10}$). Genotypic composition varies significantly along the gradient, with $KRAS^{mut}$ clone frequencies anticorrelated and $FLT3^{mut}$ correlated (Fig. 4d and Supplementary Fig. 9; Kendall rank correlation: −0.36 and 0.35, respectively, both $p < 10^{-10}$). Nevertheless, genotype composition does not fully delineate individual clonal populations, making it impossible to define distinct genotype–phenotype clusters; consequently, such gradients between the two modalities cannot be defined with conventional tools profiling a single measurement, such as flow cytometry or bulk DNA sequencing.

**FLT3 inhibitor therapy induces erythroid differentiation in a case of AML**. Our first two cases feature either a strong genotype–phenotype correlation (patient 1) or mixed genotype comprising a static immunophenotype (patient 2). Thus, for our final case, we analyzed a patient treated with gilteritinib, a FLT3 inhibitor therapy reported to promote in vivo differentiation of myeloid blasts. This treatment is thought to disperse distinct genotypes into multiple immunophenotypes, although the terminal lineage of the cells remains poorly understood[30–32]. Accordingly, we hypothesized DAb-seq permits tracking of immunophenotypic dispersal and confirmation of their terminal hematopoietic lineage. We analyzed 18,287 cells across four timepoints, beginning at diagnosis, discovering a subclone with co-mutated $DNMT3A$ and $NPM1$ (Fig. 5a and Supplementary Table 4). Following cytarabine/daunorubicin induction therapy, a fraction of $DNMT3A^{mut}$ cells remained at remission. At relapse and after treatment with the FLT3 inhibitor gilteritinib, most cells contained a 24-bp $FLT3$ internal tandem duplication (ITD), in

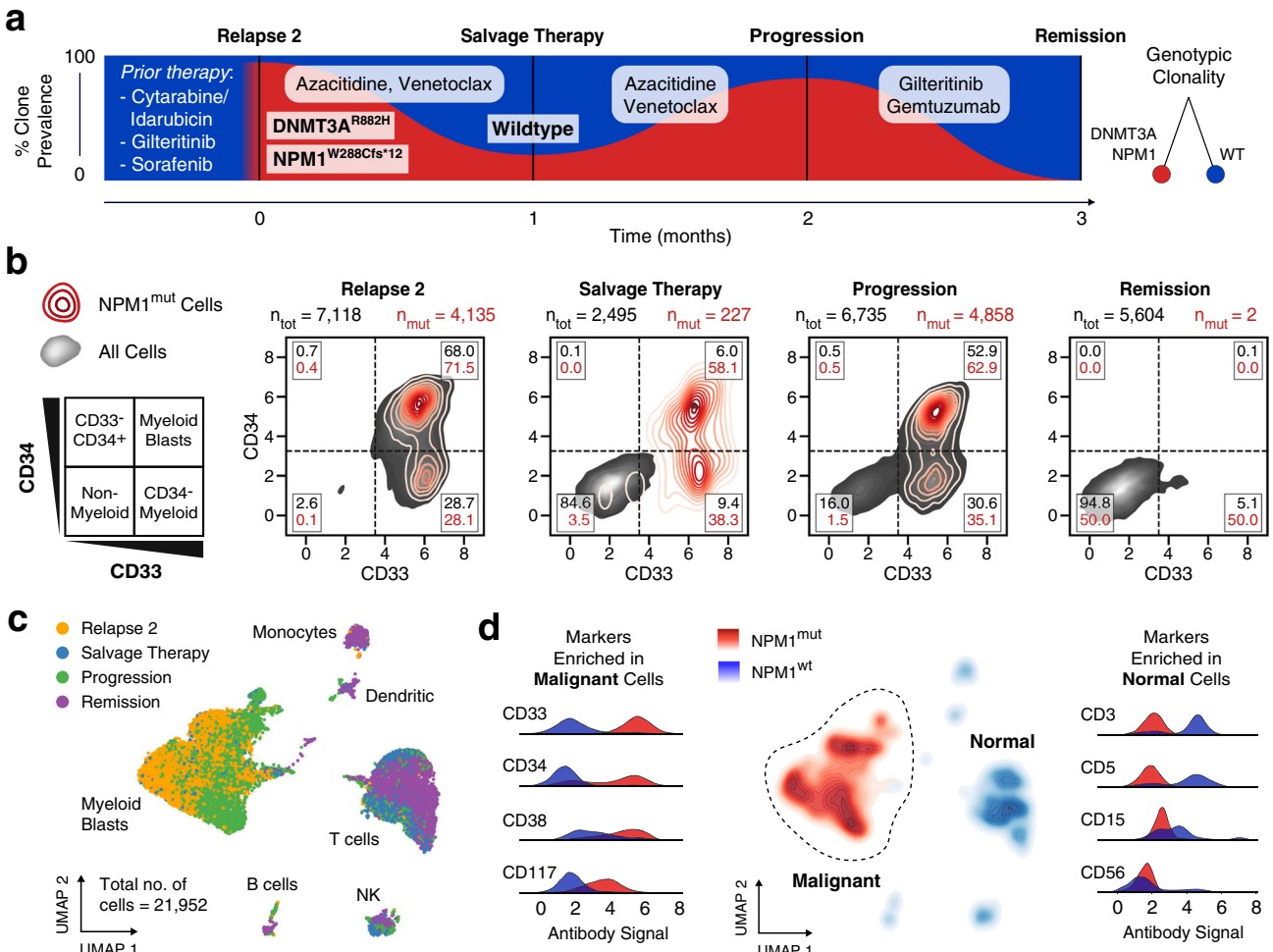

**Fig. 3 AML blasts exhibit a stable genotype and phenotype through treatment. a** DAb-seq performed on four bone marrow aspirates of a patient with AML during disease progression as indicated in the fishplot (black lines). The patient received multiple rounds of chemotherapy prior to the experiment (Supplementary Table 4). The fraction of blast cells with *NPM1* W288Cfs*12 (NPM1$^{mut}$) mutation for each sampled timepoint detected by DAb-seq are shown in red. **b** Scatter plots with kernel densities show CD33 and CD34 signal for all cells (grey) and NPM1$^{mut}$ cells (red) for each of the sampled timepoints. The percentages of normal and mutant cells within each gate are listed. Virtually gating cells highlights a persisting CD33$^+$ blast population, which is eradicated with gemtuzumab, a CD33-targeted therapy. **c** UMAP embedding based on the log-transformed and corrected antibody counts from all cells labeled by timepoint indicates that the high-dimensional immunophenotype of the blasts is stable over the sampled timepoints. **d** The genotype of each cell at the NPM1 locus is plotted as a kernel density estimate using the UMAP coordinates from **c**. Antibody signals enriched among malignant and normal populations are plotted as kernel densities using all cells and labeled by genotype. Source data are provided as a Source data file.

addition to the initial *DNMT3A* and *NPM1* mutations. The genotypic structure inferred from the single-cell data indicates a linear, branching hierarchy of sequentially acquired mutations in response to therapy. To explore the immunophenotypic features of this patient's disease, we integrate cells from all timepoints and construct a UMAP representation, using the antibody data (Fig. 5b, and Supplementary Figs. 10 and 11). We cluster this data using the Leiden method for cluster detection[33,34], and manually annotate with phenotypic labels corresponding to hematopoietic lineage from the antibody data (Fig. 5c). We identify three blast populations expressing high levels of CD33 and CD38, a monocytic population expressing CD15 and CD16, and erythroid and lymphoid clusters with elevated CD71 and CD3. As expected, samples across treatment timepoints comprise a mixture of immunophenotypically normal and blast-like cells (ARI for immunophenotype clusters and genotypes: 0.31).

Hypothesizing that different therapies should yield different genotype–phenotype coupling patterns, we sought to characterize how mutated and normal cells distribute across immunophenotypic clusters. For each timepoint, we thus label cells in UMAP space according to DNA genotype and generate density distributions of CD33 signal, a pan-myeloid marker (Fig. 5d). We also evaluate counts of phenotype cluster membership in each timepoint, subdivided by DNA genotype. At diagnosis, cells mutated at both the *DNMT3A* and *NPM1* locus reside primarily in the Blast 1 cluster (81.8% of *DNMT3A*$^{mut}$/*NPM1*$^{mut}$ cells) and express high levels of CD33. A secondary clone mutated exclusively at the *DNMT3A* locus exhibits comparable CD33 expression, and resides mainly in the Blast 1 and monocytic clusters (62.5% and 27.7% of *DNMT3A*$^{mut}$ cells, respectively). At remission, the same *DNMT3A*$^{mut}$ clone is identified but with decreased CD33 expression and a primarily monocytic immunophenotype (92.7% of *DNMT3A*$^{mut}$ cells), consistent with clonal hematopoiesis of a preleukemic clone[35,36]. A newly acquired *FLT3*-ITD clone emerges in high numbers at relapse (99.8% of genotyped cells), coinciding with a phenotypic shift of cells to the CD33$^+$ Blast 2 cluster. Following FLT3 inhibitor treatment, the same *FLT3*-ITD clone persists, but exhibits a transformed immunophenotype, as evidenced by membership of the *FLT3* clone in multiple immunophenotypic clusters. The new

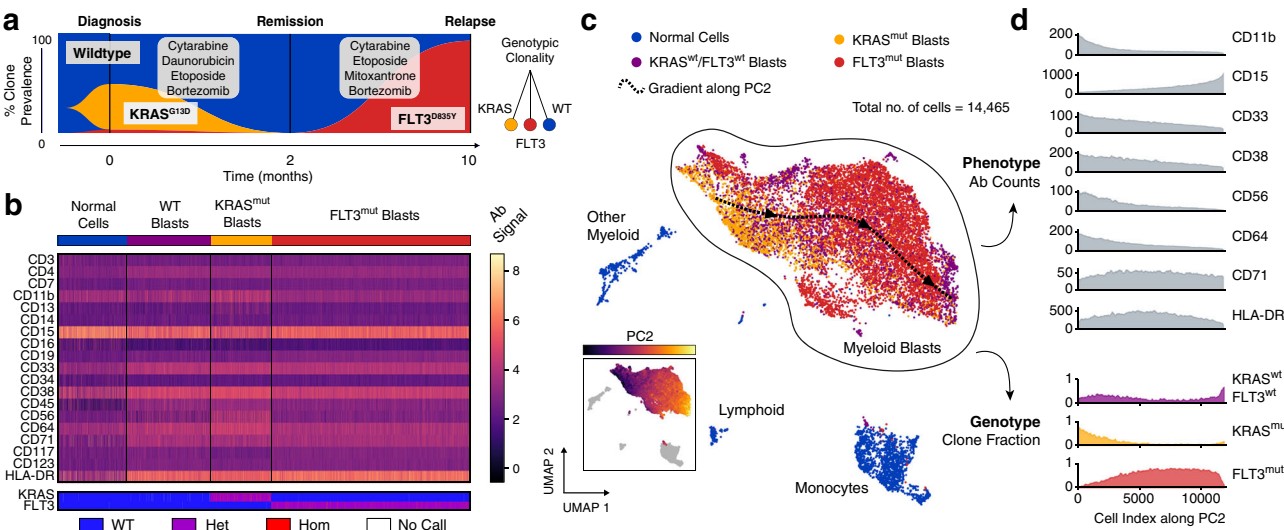

**Fig. 4 Distinct genetic subclones form an overlapping immunophenotypic continuum in a case of pediatric AML. a** Three timepoints sampled with DAb-seq during treatment comprise a mixture of independent clones (KRAS G13D heterozygous blasts, yellow; FLT3 D835Y blasts, red). The wild-type compartment contains additional cells with a blast-like immunophenotype lacking detectable mutations. **b** Heatmap of log-transformed corrected antibody counts and genotyping calls for the KRAS and FLT3 loci for each cell across all timepoints. The heatmap is grouped by genotype. Cells with wild-type genotype but blast-like immunophenotype are labeled separately. **c** UMAP embedding of all cells from all timepoints based on log-transformed corrected antibody counts. Color indicates mutation status as in **a**. The blast compartment is overlaid with a spline approximating the gradient of the second principal component of the antibody count matrix (shown in inlet figure) and indicates a gradual change in immunophenotype. **d** Moving average expression of antibodies and fraction of mutated cells sorted by the second principal component of the antibody count matrix. The overlapping phenotypic continuum between the genetically distinct blast clones is apparent. Source data are provided as a Source data file.

*FLT3*-ITD immunophenotype is primarily CD71+ erythroid (82.2% of *FLT3*-ITD cells), with minor fractions in the Blast 3 and monocytic compartments (11.1% and 4.84% of *FLT3*-ITD cells, respectively). Furthermore, the *FLT3*-ITD clone lacks uniform CD33 expression at relapse, indicating the presence of this mutation in cells outside of the myeloid compartment. The expression of CD71 is consistent with an erythroid differentiation of blasts in a case of leukemia treated with gilteritinib, which is in agreement with a recent study[32], and contrasts with a prior report of gilteritinib-induced terminal differentiation towards a myeloid fate[31]. However, a larger patient cohort will be necessary to fully characterize the extent of erythroid differentiation in gilteritinib treatment response. DAb-seq elucidates the rich dynamics of this process, and illustrates how distinct DNA genotypes can fractionate into multiple phenotypes in response to treatment.

## Discussion

Through its ability to jointly profile DNA and immunophenotype in single cells, DAb-seq captures the complexity of proteogenomic states underlying AML. Analysis of multiple samples over timepoints and treatments demonstrates potential modes of tumor evolution across different patients. In the first case, we found a robust relationship between mutant cells and a malignant phenotype. By contrast, in the second case of pediatric AML, we observed that genetically distinct populations shared an overlapping immunophenotype, demonstrating that this domain alone is insufficient for characterizing how cells are genetically programmed. In the final case, we observed the opposite scenario, in which treatment by a FLT3 inhibitor induced mutationally similar cells to disperse into different myeloid compartments, highlighting the challenge of targeting these malignant cells for eradication. Our results thus demonstrate that genotype or immunophenotype alone is insufficient to predict the evolution of proteogenomic states in AML.

DAb-seq employs targeted primers to amplify specific genomic regions and panels of antibodies. While both readouts enable massive multiplexing of queried targets, practical and economic constraints necessitate a priori knowledge of which loci and epitopes to profile. As such, DAb-seq cannot exclude the possibility that disease-relevant mutations occur beyond the sequenced loci or in immunophenotypic markers not included in the panels. In the case of pediatric AML, it is therefore impossible for us to conclude if the *FLT3wt*/*KRASwt* blast population is driven by epigenetic changes or unmapped genomic aberrations. Similarly, the discovery of a FLT3-ITD genotype after gilteritinib treatment may suggest the presence of an undetected co-occurring mutation that conveys treatment resistance. Although our genomic panel covers many common mutations at the FLT3 and RAS loci, additional amplicon targets will need to be included to capture a broader range of mutations.

DAb-seq has been successfully employed in studies of clonal evolution in myeloid malignancies[37,38]. In potential future work, DAb-seq may be used to investigate therapeutic response with high specificity. With the rise of antileukemic agents that target either cell surface markers or genetic aberrations, DAb-seq can offer information about which clones may respond to immunotherapy, small molecule therapy, or a combination. Paired with genome editing techniques, such as CRISPR, DAb-seq can be used to screen mutations in vitro at high throughput to study the phenotypic outcomes of engineered genetic variation[39]. The strength of DAb-seq is not unbiased feature discovery, as with scRNA-seq, but rather sensitive analysis across known mutational hotspots. Future studies leveraging both DNA and RNA-based single-cell multiomic tools could be designed to combine the strengths of both approaches. By performing DAb-seq and CITE-seq on the same sample, for example, it should be possible to link DNA genotyping calls and transcript counts to a common antibody space, enabling computational integration of the two modalities. The resulting data set would comprise high quality targeted genotyping information with antigen profiles and linked transcript expression, through which additional variants in the exome may be discovered.

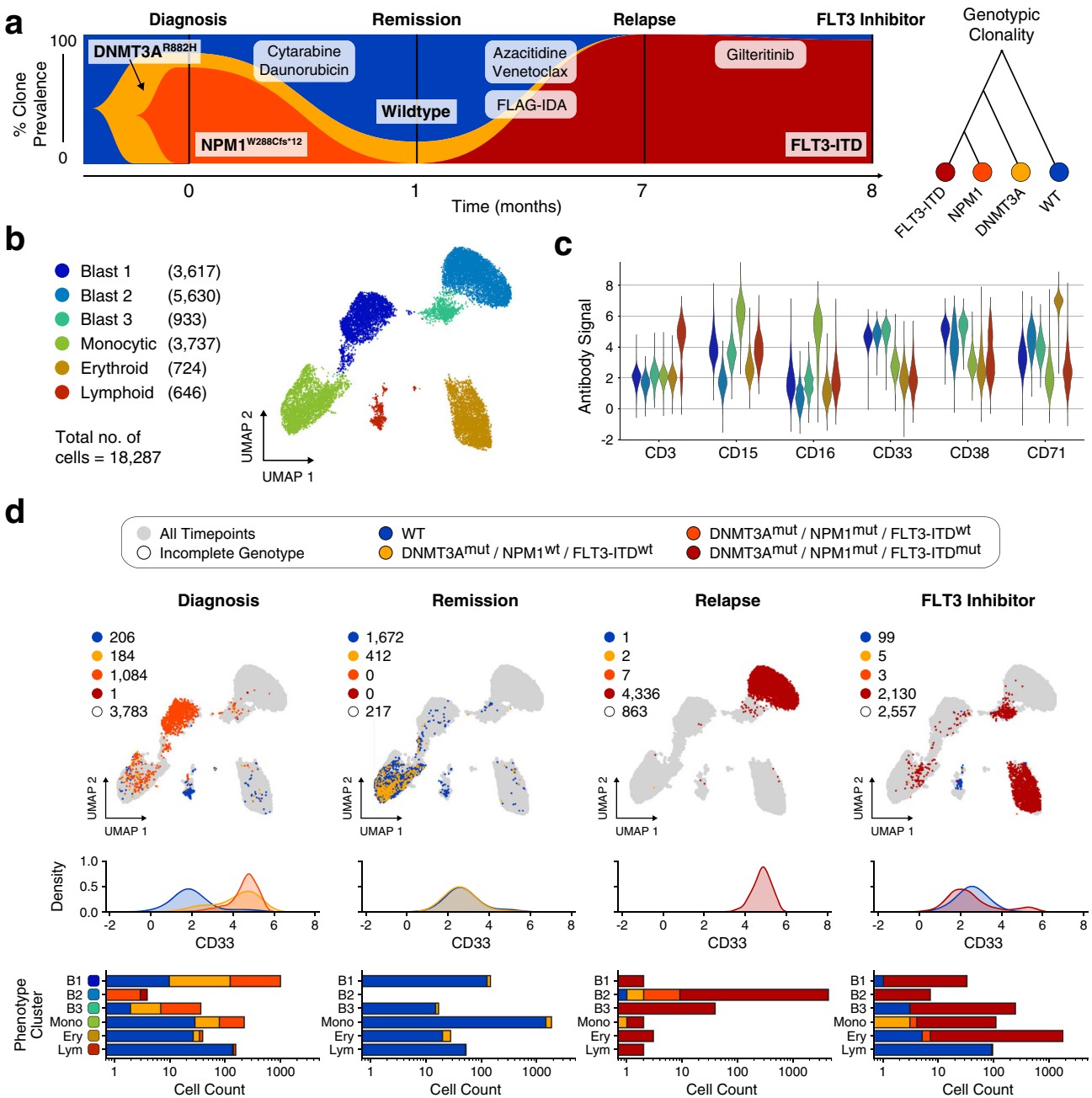

**Fig. 5 Decoupling of blast phenotype and genotype in response to FLT3 inhibitor therapy. a** Fishplot showing observed fraction of cells with distinct genetic mutations for each sampled timepoint. The co-occurrence of the three mutations in the single-cell data is consistent with a linear model of mutation accumulation. **b** UMAP embedding of all cells based on measured antibody signal. The cells segregate into six distinct phenotypic clusters with multiple blast compartments. **c** Average expression of each cell cluster for a selection of markers. **d** Top row: Same UMAP embedding as in **b** given as grey outline. For each sampled timepoint, observed cells are plotted and colored according to the detected genotype. Blasts distribute among multiple phenotypic compartments in the final timepoint following FLT3 inhibitor treatment. Middle row: kernel density plot of the CD33 antibody signal resolved by timepoint and genotype. Cells from genotypic compartments with less than ten cells per timepoint are not plotted. Bottom row: bar chart depicting genotypic composition of each phenotypic cluster in **b** resolved by timepoint. Source data are provided as a Source data file.

## Methods

A detailed step-by-step protocol for DAb-seq has been submitted to the protocols. io repository[40]. The cost analysis for a DAb-seq experiment is provided in Supplementary Table 7.

**Conjugation of antibodies to oligonucleotide barcodes**. Monoclonal antibodies were conjugated to azide-modified oligonucleotides using a copper-free click chemistry reaction[41]. Monoclonal antibodies were resuspended to 100 μg in 100 μL PBS. See Supplementary Table 2 for a complete list of antibodies and oligonucleotide barcode sequences. Antibodies were incubated with DBCO-PEG5-NHS

Ester linker (Click Chemistry Tools, cat. no. A102P) at a 4:1 molar ratio linker: antibody for 2 h at room temperature. Following incubation, the antibody-linker solution was washed once in a 50 kDa cellulose spin filter (Millipore Sigma, cat. no. UFC505024). DNA oligonucleotides with a 5′ azide modification (Integrated DNA Technologies) were reconstituted in water and added to the washed antibodies at a 2.5:1 molar ratio oligonucleotide:antibody. Following a 16 h incubation, the conjugated antibodies were washed three times in a 50 kDa filter to remove unreacted oligonucleotides. All antibody conjugates were run on a Bioanalyzer Protein 230 electrophoresis chip (Agilent Technologies, cat. no. 5067-1517) to verify successful conjugation.

**Cell culture and PBMC processing for control experiments**. The following three cell lines were used in the initial control experiment: Raji (ATCC, CCL-86), Jurkat (ATCC, TIB-152), and K562 (ATCC, CCL-243). Cells were cultured under the supplier's recommended conditions. PBMCs from a single healthy donor were sourced commercially (iXCells Biotechnologies, cat. no. 10HU-003) and stored at −80 °C until use. Prior to staining, the cultured cell lines and PBMCs were washed once in PBS with 5% fetal bovine serum (FBS; Thermo Fisher, cat. no. 10082147). For the control experiment, the three cell lines were combined at an equal ratio.

**Collection of patient samples**. Patients included in this study were treated at the University of California, San Francisco (UCSF), and peripheral blood or bone marrow was stored in the UCSF tumor bank. Samples were processed immediately after collection to isolate mononuclear cells. Sample collection was in accordance with the Declaration of Helsinki under tissue banking protocols approved by the UCSF Committee on Human Research. Written informed consent was obtained from all patients. A summary of clinical histories and flow cytometry reports for each patient is available in Supplementary Tables 4 and 5, respectively.

**Thawing patient samples**. A protocol was optimized to maximize recovery of viable cells from patient samples. Cryovials containing patient tissue (peripheral blood or bone marrow aspirate) were warmed by hand and carefully transferred dropwise to a 50 mL tube containing 40 mL of cold DMEM media (Thermo Fisher, cat. no. 11995040) with 20% FBS and 2 mM EDTA. The tube was centrifuged at $200 \times g$ at 4 °C for 7 min with no brake. The supernatant was discarded, and the cells were resuspended in 10 mL of warmed RPMI-1640 media (Thermo Fisher, cat. no. A1049101) with 10% FBS. The solution was strained through a 70 μm cell strainer (Corning, cat. no. 431751) to remove any large cell aggregates, and the tube was centrifuged a second time at $200 \times g$ at 4 °C for 5 min with low brake. The supernatant was discarded, and the cells were resuspended in PBS with 5% FBS for staining.

**Cell staining using oligonucleotide-conjugated antibodies**. For each sample, 2 million cells were added to a 5 mL DNA LoBind tube (Eppendorf, cat. no. 0030108310), centrifuged at $400 \times g$ for 4 min, and resuspended in 180 μL PBS with 5% FBS. Cells were blocked for 10 min on ice following addition of 10 μL Fc blocking solution (BioLegend, cat. no. 422301), 4 μL of a 1% dextran sulfate solution (Research Products International, cat. no. D20020), and 4 μL of 10 mg/mL salmon sperm DNA (Invitrogen, cat. no. 15632011). Cells were stained for 30 min on ice with 0.5 μg of each conjugated antibody. After incubation, five rounds of washing were performed to remove excess antibody. For each wash, 5 mL PBS with 5% FBS was added to the tube and centrifuged at $400 \times g$ for 4 min. Stained cells were resuspended in Mission Bio cell buffer at a final concentration of 3 M/mL prior to microfluidic encapsulation.

**Microfluidic single-cell DNA genotyping and antibody capture**. A commercial single-cell DNA genotyping platform (Mission Bio, Tapestri) was used to perform microfluidic encapsulation, lysis, and barcoding according to the manufacturer's protocol for the AML V1 panel. Where noted, modifications were made to enable co-capture of oligonucleotide-labeled antibodies. Stained cells were loaded into a microfluidic cartridge and co-encapsulated into droplets with a lysis buffer containing protease and mild detergent. Droplets were incubated in a thermal cycler for 1 h at 50 °C to digest all cellular proteins, followed by 10 min at 80 °C to heat inactivate the protease. To enable antibody capture during the barcoding stage, the antibody tags were designed with 3′ complementarity to one of the *RUNX1* gene forward primers and the corresponding reverse primer was omitted from the reverse primer pool. Supplementary Table 1 lists the sequences of the forward and reverse primers in the DNA panel. Lysed cells in droplets were transferred to the barcoding module of the microfluidic cartridge in addition to polymerase mix, the modified reverse primer pool, barcoded hydrogel beads, and oil for droplet generation. The droplets were placed under a UV lamp (Analytik Jena, Blak-Ray XX15L) for 8 min to cleave the single-stranded PCR primers containing unique cell barcodes from the hydrogel beads. To amplify DNA targets and capture antibody tags, droplets were thermal cycled using the following program: 95 °C for 10 m; 20 cycles of 95 °C for 30 s, 72 °C for 10 s, 61 °C for 4 min, 72 °C for 30 s; 72 °C for 2 min; 4 °C hold.

**Single-cell DNA amplicon and antibody tag sequencing library preparation**. Recovery and cleanup of single-cell libraries proceeded according to the Mission Bio V1 protocol with additional modifications for antibody library preparation. The eight PCR tubes containing barcoded droplets were pooled as pairs and treated with Mission Bio Extraction Agent. Water was added to each tube and the aqueous fraction transferred to a new 1.5 mL DNA LoBind tube. Ampure XP beads (Beckman Coulter, cat. no. A63881) were added at a 0.75× volume ratio beads:PCR product for size selection. The supernatant from the size selection step, containing library fragments shorter than ~200 bp, was retained and used for antibody library preparation, while the remaining beads with bound DNA panel library fragments were washed twice with 80% EtOH and eluted in 30 μL water. A biotinylated capture oligonucleotide (/5Biosg/GGCTTGTTGTGATTCGACGA/3C6/, Integrated DNA Technologies) complementary to the 5′ end of the antibody tags was added to the retained supernatant to a final concentration of 0.6 μM. The

supernatant-probe solution was heated to 95 °C for 5 min to denature the PCR product, then snap-cooled on ice for probe hybridization. A total of 10 μL of streptavidin beads (Thermo Fisher, cat. no. 65001) were washed according to the manufacturer's protocol and added to each tube of PCR product. Following a 15 min incubation at room temperature, the beads were isolated by magnetic separation, washed two times in PBS, and resuspended in 30 μL water. PCR was performed on the purified DNA panel and antibody tags to produce sequencing libraries. For each tube of purified DNA panel, 50 μL reactions were prepared containing 4 ng of barcoded product in 15 μL water, 25 μL Mission Bio Library Mix, and 5 μL each of custom P5 and Nextera P7 primers (N7XX), both at 4 μM stock concentration. The reactions were thermal cycled using the following program: 95 °C for 3 min; ten cycles of 98 °C for 20 s, 62 °C for 20 s, 72 °C for 45 s; 72 °C for 2 min; 4 °C hold. For each tube of purified antibody tags, identical reactions were prepared, instead using 15 μL bead-bound template, 5 μL antibody tag-specific P7 primer at 4 μM, and 20 cycles of amplification. See Supplementary Table 6 for a complete listing of custom library preparation primers. Following amplification, both the DNA panel and antibody tag libraries were cleaned with 0.7× Ampure XP beads and eluted in 12 μL water.

**Next-generation sequencing**. All DNA panel and antibody tag libraries were run on a Bioanalyzer High Sensitivity DNA electrophoresis chip (Agilent Technologies, cat. no. 5067-4626) to verify complete removal of primer-dimer products. Libraries were quantified by fluorometer (Qubit 3.0, Invitrogen) and sequenced on Illumina next-generation sequencing platforms with a 20% spike-in of PhiX control DNA (Illumina, cat. no. FC-110-3001). All sequencing runs used a dual-index configuration and a custom Read 1 primer (5′ GCCTGTCCGCGGAAGCAGTGGTAT CAACGCAGAGTAG 3′, Integrated DNA Technologies). The three-cell control sample was sequenced on an Illumina MiSeq using a v2 300-cycle kit in $2 \times 150$ bp paired-end mode (Illumina, cat. no. MS-102-2002). For the patient samples, DNA panel and antibody tag libraries were sequenced separately for cost-effectiveness. DNA panels were sequenced with an Illumina NovaSeq 6000 SP 300-cycle Kit (Illumina, cat. no. 20027465) in $2 \times 150$ bp paired-end mode. Antibody tag libraries were sequenced with an Illumina NextSeq 550 75-cycle High Output Kit (Illumina, cat. no. 20024906) in paired-end mode, using 38 cycles for Read 1 and 39 cycles for Read 2.

**Bioinformatic pipeline for single-cell DNA genotyping and antibody tag counting**. Sequencing data were processed using a custom pipeline available on GitHub (see "Code availability" section). For all reads, combinatorial cell barcodes were parsed from Read 1, using cutadapt (v2.4) and matched to a barcode whitelist. Barcode sequences within a Hamming distance of 1 from a whitelist barcode were corrected.

For the DNA genotyping libraries, reads with valid barcodes were trimmed with cutadapt to remove 5′ and 3′ adapter sequences and demultiplexed into single-cell FASTQ files, using the script demuxbyname.sh from the BBMap package (v.38.57). Valid cell barcodes were selected using the inflection point of the cell rank plot in addition to the requirement that 60% of DNA intervals were covered by a minimum of eight reads. FASTQ files for valid cells were aligned to the hg19 build of the human genome reference using bowtie2 (v2.3.4.1). The single-cell alignments in BAM format were filtered (properly mapped, mapping quality > 2, primary alignment), sorted, and indexed with samtools (v1.8). GVCF files were produced for all cells using HaplotypeCaller from the GATK suite (v.4.1.3.0). Joint genotyping was performed on all genomic intervals in parallel (excluding primer regions), using GATK GenotypeGVCFs. For longitudinal patient samples, cells from all timepoints were joint genotyped as a multi-sample cohort. Genotyped intervals from all cells were combined into a single variant call format (VCF) file, and multiallelic records were split and left-aligned using bcftools (v1.9). Variants were annotated with ClinVar metadata (v.20190805) and SnpEff functional impact predictions (v4.3t). Variant records for all cells were exported to HDF5 format using a condensed representation of the genotyping calls (0: wild type; 1: heterozygous alternate; 2: homozygous alternate; 3: no call).

The antibody tag libraries were processed identically for cell barcode demultiplexing. For reads with valid cell barcodes, 8 bp antibody barcodes and 10 bp unique molecular identifiers (UMIs) were extracted from Read 2, using cutadapt with the requirement that all UMI bases had a minimum quality score of 20. Antibody barcode sequences within a Hamming distance of 1 from known antibody barcodes were corrected. UMI sequences were grouped by cell and antibody and counted, using the UMI-tools package[42] (v.0.5.3, "adjacency" method). UMI counts of antibodies for each cell barcode were exported in tabular format for further analysis.

**Cell and genotype filtering**. Cell barcodes were additionally filtered according to antibody counts. Valid barcode groups were required to have a minimum of 100 antibody UMIs by the adjacency counting method and a maximum IgG1 count no greater than five times the median IgG1 count of the associated DAb-seq experiment. For each valid cell barcode, all variants were filtered according to the quality and sequence depth reported by GATK. Genotyping calls were required to have a minimum quality of 30 and total depth of 10; variant entries below these thresholds were marked as "no call" and excluded from analyses.

**Antibody-based embedding and clustering**. To correct for technical effects in the raw antibody counts and batch variability between experiments from the same patient but different timepoints, a linear regression over all cells from the same patient was performed. Specifically, to all entries $c_{ij}$ of the UMI corrected antibody count matrix $c$, where $i$ is the cell index and $j$ the antibody index, one pseudocount was added and the matrix was log-transformed. A matrix of quality metrics $q$ with cells as rows and four columns (total antibody reads, total antibody counts after UMI correction, IgG1 count, and total amplicon reads) was log-transformed, column-wise normalized, and mean-centered. A singular value decomposition was performed on the transformed matrix $q$ and the left-singular vectors retained as design matrix. Each column vector $c_j$ was then regressed with either the first three, two, or one left-singular vectors, for patient samples, PMBC or cell lines, respectively, as regressors. The vector of residuals $u_j$ is then the corrected antibody signal of antibody $j$ (Supplementary Figs. 12 and 13).

A UMAP embedding in two dimensions of the corrected antibody signal was done in Python 2.7, using the umap-learn[28] (v0.3.10) and scanpy[43] (v.1.4.4.post1) packages, with the minimum distance parameter set to 0.1 for the pediatric patient and 0.2 for all other samples and default parameters otherwise. To construct the underlying nearest neighbor graph from the corrected antibody count matrix, 15 or 16 nearest neighbors based on the first 16 to all principal components were used. The scanpy implementation of the Leiden algorithm[33] with resolution set to 0.1 for the three-cell line experiment and 1 otherwise was used to assign cells to phenotypic compartments.

For the gradient analysis of the pediatric patient with AML (Fig. 4), only cells belonging to Leiden communities with blast phenotype were retained and the singular value decomposition of the remaining rows of $u$ was calculated. Cells were then ordered by their value of the second left-singular vector. Antibody counts and genotype fractions along the gradient were averaged with a moving window of 200 cells. Similarly, the average position of the cells in the two-dimensional UMAP embedding was estimated by smoothing $x$ and $y$ coordinates with a moving window of the same length. A third-order spline was placed through the smoothed cell position to indicate the orientation of the gradient in the UMAP embedding.

**Clustering statistics calculations**. To calculate the ARI (corrected for matches by chance), the implementation in the Python package scikit-learn library v.0.21.3 was used. Genotypic clusters were defined as the combined variant call for the relevant loci within a sample. For example, using two relevant loci with four variant calls (wild type, heterozygous, homozygous alternate, and no call) as in Fig. 3, 16 genotype assignments may exist. For the patient samples, heterozygous and homozygous alternate calls were classified together to reduce noise stemming from allele dropout. Phenotypic clusters correspond to the Leiden community assignment as shown in the figures. The same clusters were used in comparisons based on the uncertainty coefficient, defined as: $U(X|Y) = I(X;Y)/H(X)$, where $I(X;Y)$ is the mutual information of the distributions $X$ and $Y$ and $H(X)$ the entropy in $X$. The uncertainty coefficient is a measure of what fraction of information in $X$ is predictable given $Y$.

**Reporting summary**. Further information on research design is available in the Nature Research Reporting Summary linked to this article.

## Data availability

Sequence data that support the findings of this study have been deposited in the Sequencing Read Archive under BioProject accession number "PRJNA602320". The authors declare that all other data supporting the findings of this study are available within the paper and its supplementary information files. Source data are provided with this paper.

## Code availability

The DAb-seq bioinformatic pipeline is available on GitHub at https://github.com/AbateLab/DAb-seq[44].

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

## Acknowledgements
This work was supported by the Chan Zuckerberg Biohub, the National Science Foundation CAREER Award (Award Number DBI-1253293), and the National Institutes of Health (Award Numbers 2R01EB019453 and 1DP2AR068129). C.L.D. is supported by a Swiss National Science Foundation fellowship (Grant No. 183853). C.A.C.P. is supported by Alex's Lemonade Stand Young Investigator Award, the Conquer Cancer ASCO Young Investigator Award, and by the National Center for Advancing Translational Sciences of the NIH. C.C.S. is supported by a Research Scholar Award from the American Cancer Society and is the Damon Runyon-Richard Lumsden Foundation Clinical Investigator supported (in part) by the Damon Runyon Cancer Research Foundation (CI-99-18). Research contents are solely the responsibility of the authors and do not necessarily represent the official views of the NIH. We thank Aik Ooi at Mission Bio for his help with antibody conjugations and Mission Bio for donating reagents.

## Author contributions
B.D. and C.L.D. performed the experiments, sequenced the samples, analyzed the data, and wrote the initial draft of the manuscript. C.A.C.P. assisted with patient sample processing. H.N.V. and A.R.A. revised the manuscript. C.C.S. treated the patients and obtained samples. All authors read, reviewed, and approved the manuscript.

## Competing interests
A.R.A. is a co-founder and shareholder of Mission Bio, Inc. D.R. is an employee of Mission Bio, Inc. All other authors declare no competing interests.
