## [Peer Review File · Nature Communications]

Reviewers' Comments:

Reviewer #1:

Remarks to the Author:

In this paper the authors develop a protocol to simultaneously genotype and immunophenotype single cells. A method that combines these two single-cell measurements in this way has not been published to my knowledge and is highly useful, for example in the context of characterizing AML blasts as done in this paper.

I think overall this is a highly useful paper, well written and illustrated.

I would have a few suggestions to make this paper even more useful

1) I think a potential worry is the formation of chimeras when libraries are amplified leading to wrong assignments of cell barcodes and genotypes and phenotypes (although the former would be of more concern). This phenomenon is well known in rRNA amplicon libraries and is also more and more addressed in scRNA-seq libraries (e.g. Yang, Shiyi, Sean E. Corbett, Yusuke Koga, Zhe Wang, W. Evan Johnson, Masanao Yajima, and Joshua D. Campbell. 2020. "Decontamination of Ambient RNA in Single-Cell RNA-Seq with DecontX." *Genome Biology* 21 (1): 57.) I think it would be important to address this point quantitatively by estimating the fraction of chimeras and how this would impact the assignments of single cell genotypes in relation to the number of loci that differ between cells.

2) I think the authors should make the costs of the method transparent at least in the supplement (for set-up and per run) to help the reader to gauge the useability in their own context

3) It would be important to publish a detailed protocol. As the authors are from Biohub, I guess protocols.io would be an obvious choice.

Minor comments

P1.l43 this sentence does not seem logical to me

p.2.l62 "while their counts relate phenotype" rather "relate to.."? Not sure what is meant

Reviewer #2:

Remarks to the Author:

In this study Demaree B and co-authors described a new developed method, Dab-seq, for simultaneous capture of DNA genotype and cell surface phenotype from single cells, enabling direct profiling of proteogenomic states in thousands of cells. This approach was applied to the analysis of clonal evolution in response to therapy in three different AML patients who received multiple treatment regimens including both conventional chemotherapy and/or targeted therapies. Analysis of multiple samples overtime showed different trajectories of tumor evolution across different patients, demonstrating that genotype or immunophenotype alone is insufficient to predict the evolution of proteogenomic states. The power of single-cell DNA-sequencing for predicting relapse has been demonstrated in previously published study by using a similar approach but limited to DNA profiling. The simultaneous profiling of both cell surface markers and genotype changes has important clinical and therapeutic implications.

Although the study is methodologically well performed with appropriate negative and positive controls and robust data analysis, it may benefit of some improvements in patient and molecular data description.

Major comments

Although it seems that the main message of this paper is more methodological rather than clinical, key patient characteristics should be described to interpret the results. Please provide demographic and clinical (age, karyotype, WHO 2017 classification, ELN risk) characteristics for the three AML patients analyzed in this study.

I wonder if the authors performed bulk DNA sequencing (whole genome sequencing or whole exome sequencing) on samples at diagnosis. It is not clear how many mutations per samples were covered from the amplicon panel used for single-cell sequencing analysis. For example, it is not mentioned whether the samples with NPM1 negative had a recurrent gene fusion or other alterations in addition to those here screened.

The major concern is regarding the patient number three (Fig. 5) with co-mutated DNMT3A and NPM1 at diagnosis and with arising FLT3-ITD following treatment with the FLT3 inhibitor gilteritinib. Gilteritinib has been shown to be active against a wide range of clinically relevant activating FLT3 mutations, including ITD. How do the authors explain the emergence of FLT3-ITD mutation? Several studies, including some performed by single-cell DNA sequencing, identified treatment-emergent mutations that activate RAS/MAPK pathway signaling, most commonly in NRAS or KRAS as mechanism of resistance to FLT3 inhibition. Did the authors here investigate RAS pathway mutations?

By correlating immunophenotype with distribution of mutant and normal cells overtime, the authors concluded that the new FLT3-ITD immunophenotype is primarily erythroid. However, except for the expression of CD71 (that is not exclusive of erythroid cells), it is not clear how the authors defined erythroid cells. This is crucial for the conclusion and more immunophenotype and molecular data should be definitively investigated and provided. The conclusion on page 11, lines 256-257, "these findings support the model of terminal erythroid differentiation of blasts in a case of leukemia treated with gilteritinib" appears very strong and since it results from only one case it should be toned-down.

Minor

Page 9, line 221: please specify "a fraction"

Reviewer #3:

Remarks to the Author:

The manuscript by Demaree et al presents DAb-seq, a very useful technique for joint profiling of point mutations and surface proteins in single cells. Built upon previous techniques from the lab, the study is well conducted as expected. I only have a few minor comments.

1. The manuscript can benefit from some discussion regarding possible application in clinical and basic research.
2. The authors have pointed out the "targetd analysis" limitation, which could be compensated by scRNA-seq. How can the method be further strengthened by some companion RNA-seq analysis (bulk and/or single cell).

Joint profiling of DNA and proteins in single cells to dissect genotype-phenotype associations in leukemia

Demaree and Delley, et al.

NCOMMS-20-36979

Point-by-point responses to reviewer comments are provided below.

Reviewer #1

Comment 1: In this paper the authors develop a protocol to simultaneously genotype and immunophenotype single cells. A method that combines these two single-cell measurements in this way has not been published to my knowledge and is highly useful, for example in the context of characterizing AML blasts as done in this paper. I think overall this is a highly useful paper, well written and illustrated.

Response: We thank the reviewer for critically evaluating our manuscript and for the helpful suggestions.

Comment 2: I would have a few suggestions to make this paper even more useful: I think a potential worry is the formation of chimeras when libraries are amplified leading to wrong assignments of cell barcodes and genotypes and phenotypes (although the former would be of more concern). This phenomenon is well known in rRNA amplicon libraries and is also more and more addressed in scRNA-seq libraries (e.g. Yang, Shiyi, Sean E. Corbett, Yusuke Koga, Zhe Wang, W. Evan Johnson, Masanao Yajima, and Joshua D. Campbell. 2020. "Decontamination of Ambient RNA in Single-Cell RNA-Seq with DecontX." *Genome Biology* 21 (1): 57.) I think it would be important to address this point quantitatively by estimating the fraction of chimeras and how this would impact the assignments of single cell genotypes in relation to the number of loci that differ between cells.

Response: We agree that a critical examination of the noise is important, particularly in situations where minute quantities of nucleic acids require many PCR cycles to amplify. We believe DAb-seq to be robust against these effects for the reasons explained below.

As the reviewer correctly points out, the antibody counts are not low abundance which is apparent from the fact that we sequence few PCR duplicates (that is, most unique molecular identifiers are observed only once, Supplementary Figure 12B) and observe a signal to noise ratio of 10-100:1 for individual antibodies (compare Supplementary Figure 12B and C). Chimera formation is one possible source for these background counts as are free antibodies in solution. We think that the latter case might in fact be the dominant cause, because chimera formation should result in an increased noise floor for antibodies which are highly prevalent in a sample

such as CD5 or CD13 in PBMC. However, we observe a noise floor comparable to the isotype control IgG1, indicating that the noise counts do not scale with expression (Supplementary Figure 12B).

For genotype calls, in contrast to single-cell RNA experiments, where a background of ambient RNA in the cell suspension (e.g. from damaged cells) contributes to the noise, chromosomal DNA should be less prone than RNA to escape individual cells and contribute to ambient noise.

We also expect the DAb-seq genotype readout to be robust against chimera formation/barcode hopping because all genomic targets are amplified in droplets for 20 cycles prior to the library preparation. This compartmentalization prevents barcode exchange while amplifying template copy number up to ~1 million-fold. There is a small likelihood of chimera formation in the subsequent library preparation PCR (10 cycles), but any individual crossover event will not amplify to more than ~0.1% of the final amplicon pool ($2^{10}/2^{20}$ library amplification / preamplification). We discard such low frequency events by the implemented variant filtering criteria to assign a variant call (see Methods).

In addition, our control experiment using cell lines (Figure 2d) shows minimal mixing between cell genotypes and phenotypes, suggesting that chimera formation - and other potential noise sources such as cell doublet encapsulation and droplet merging - do not contribute strongly to the detected single-cell genotypes. To quantify these sources of noise, we have counted the number of false genotyping calls in the experiment using the antibody data as the ground truth for cell identity. In cases where cell mixing has occurred - whether due to cell doublets, droplet merger, or PCR recombination - we expect a discordance between the genotypes and phenotypes in the cell line experiment. To exclude stochastic incidents of allele drop out, we consider only cases where a known wildtype locus has been erroneously called as a heterozygous or homozygous mutation, which we do not expect to observe in clonal cell lines. The five such cases shown in Figure 2d are listed in “Supplementary Table 8: Genotyping metrics in cell line experiment.” We find that the average percentage of incorrect calls across comparisons is 6.6%, which can be explained by the rate of cell multipliers on the Tapestry platform, reported by Mission Bio to be <8% (<https://support.missionbio.com>). Therefore, we do not expect chimera formation to contribute strongly to experimental noise.

To address the reviewer’s comments in the manuscript, we have added Supplemental Table 8 and amended the Results section with a reference to this data.

Comment 3: I think the authors should make the costs of the method transparent at least in the supplement (for set-up and per run) to help the reader to gauge the useability in their own context.

Response: We thank the reviewer for this suggestion. To address the comment, we have amended the manuscript as requested and added “Supplementary Table 7: Cost analysis of DAb-seq” with cost estimates, both on an absolute and per-experiment basis. The table is referenced under the Methods section.

Comment 4: It would be important to publish a detailed protocol. As the authors are from Biohub, I guess protocols.io would be an obvious choice.

Response: We agree that a detailed protocol will be helpful to others in implementing DAb-seq. To address the reviewer’s comment, we have done as requested and prepared a detailed protocol and posted it on the protocols.io platform, available at: [dx.doi.org/10.17504/protocols.io.bn4ymgxw](https://doi.org/10.17504/protocols.io.bn4ymgxw). We have also updated the manuscript with a link to the protocol (under “Methods”).

Comment 5: P1.l43 this sentence does not seem logical to me p.2.l62 “while their counts relate phenotype” rather “relate to..”? Not sure what is meant

Response: We have clarified these sentences in the manuscript.

Reviewer #2

Comment 1: In this study Demaree B and co-authors described a new developed method, Dab-seq, for simultaneous capture of DNA genotype and cell surface phenotype from single cells, enabling direct profiling of proteogenomic states in thousands of cells. This approach was applied to the analysis of clonal evolution in response to therapy in three different AML patients who received multiple treatment regimens including both conventional chemotherapy and/or targeted therapies. Analysis of multiple samples overtime showed different trajectories of tumor evolution across different patients, demonstrating that genotype or immunophenotype alone is insufficient to predict the evolution of proteogenomic states. The power of single-cell DNA-sequencing for predicting relapse has been demonstrated in previously published study by using a similar approach but limited to DNA profiling. The simultaneous profiling of both cell surface markers and genotype changes has important clinical and therapeutic implications. Although the study is methodologically well performed with appropriate negative and positive controls and robust data analysis, it may benefit of some improvements in patient and molecular data description.

Response: We thank the reviewer for their time and help in reviewing our manuscript.

Comment 2: Although it seems that the main message of this paper is more methodological rather than clinical, key patient characteristics should be described to interpret the results. Please

provide demographic and clinical (age, karyotype, WHO 2017 classification, ELN risk) characteristics for the three AML patients analyzed in this study.

Response: We added the requested information to the manuscript in “Supplementary Table 4: Patient information and treatment histories.”

Comment 3: I wonder if the authors performed bulk DNA sequencing (whole genome sequencing or whole exome sequencing) on samples at diagnosis. It is not clear how many mutations per samples were covered from the amplicon panel used for single-cell sequencing analysis. For example, it is not mentioned whether the samples with NPM1 negative had a recurrent gene fusion or other alterations in addition to those here screened.

Response: NGS and qPCR tests were performed at the time of diagnosis, however, some patients were diagnosed prior to the routine use of clinical NGS. We have added the available information to “Supplementary Table 4: Patient information and treatment histories.”

Comment 4: The major concern is regarding the patient number three (Fig. 5) with co-mutated DNMT3A and NPM1 at diagnosis and with arising FLT3-ITD following treatment with the FLT3 inhibitor gilteritinib. Gilteritinib has been shown to be active against a wide range of clinically relevant activating FLT3 mutations, including ITD. How do the authors explain the emergence of FLT3-ITD mutation? Several studies, including some performed by single-cell DNA sequencing, identified treatment-emergent mutations that activate RAS/MAPK pathway signaling, most commonly in NRAS or KRAS as mechanism of resistance to FLT3 inhibition. Did the authors here investigate RAS pathway mutations?

Response: We agree that the predominance of FLT3-ITD harboring clone could suggest the existence of an undetected co-occurring resistance mutation. Alternatively, these cells could be residually present from prior treatment and may eventually die. Our panel covers the most common resistance mutations in the FLT3 and RAS pathway; these locations were detected consistently (Supplementary Figure 1) but did not exhibit pathological mutations. To address the reviewer’s comment, we have amended the manuscript to discuss these possibilities.

Comment 5: By correlating immunophenotype with distribution of mutant and normal cells overtime, the authors concluded that the new FLT3-ITD immunophenotype is primarily erythroid. However, except for the expression of CD71 (that is not exclusive of erythroid cells), it is not clear how the authors defined erythroid cells. This is crucial for the conclusion and more immunophenotype and molecular data should be definitively investigated and provided. The conclusion on page 11, lines 256-257, “these findings support the model of terminal erythroid differentiation of blasts in a case of leukemia treated with gilteritinib” appears very strong and since it results from only one case it should be toned-down.

Response: We thank the reviewer for this suggestion. To address the reviewer's comment, we have amended the manuscript to tone-down this statement.

Comment 6: Page 9, line 221: please specify "a fraction"

Response: We added the specific number to the manuscript.

Reviewer #3

Comment 1: The manuscript by Demaree et al presents DAb-seq, a very useful technique for joint profiling of point mutations and surface proteins in single cells. Built upon previous techniques from the lab, the study is well conducted as expected. I only have a few minor comments.

Response: We thank the reviewer for their time evaluating our manuscript.

Comment 2: The manuscript can benefit from some discussion regarding possible application in clinical and basic research.

Response: We thank the reviewer for this suggestion. We have amended the manuscript with a discussion of potential clinical and basic research applications of this work.

Comment 3: The authors have pointed out the "targetd analysis" limitation, which could be compensated by scRNA-seq. How can the method be further strengthened by some companion RNA-seq analysis (bulk and/or single cell).

Response: We agree with the reviewer that a DAb-seq experiment can be nicely complemented with other approaches. For instance, it should be possible to apply bulk whole exome sequencing or bulk RNA sequencing to the same patient samples and exploit the genotype-phenotype associations obtained from a DAb-seq experiment to guide inference of expanded cell signatures. Similar approaches that combine single-cell RNA with bulk RNA data to counter transcript dropouts in single cells have been recently described (e.g. Zhu et al., "A unified statistical framework for single cell and bulk RNA sequencing data", Ann. Appl. Stat. 2018; Peng et al., "SCRABBLE: single-cell RNA-seq imputation constrained by bulk RNA-seq data", Genome Biology 2019).

Another possible approach would be to pair a DAb-seq experiment with a single-cell RNA-seq (scRNA-seq) experiment, which could potentially allow accurate linking of single-cell genotypes with transcript profiles and protein expression. As we discuss in the manuscript, single-cell RNA

experiments have the intrinsic benefit that variants can be called from the entire exome, which may yield genotype information from outside of the amplicons covered by any particular DAb-seq panel. Transcript profiles from scRNA-seq can also be used to explore phenotypic heterogeneity in cell compartments that are not well covered with a given set of antibodies. These capabilities come with the caveat that genotyping from transcripts yields very sparse data, with many variants only called in ~1-10% of cells, and with no allelic resolution in most cases (e.g. A. Nam, K. Kim, R Chaligne et al., "Somatic mutations and cell identity linked by Genotyping of Transcriptomes", Nature 2019). Furthermore, transcript profiles as phenotypic readouts are less easy to interpret than immunophenotypes, where decades of literature exists.

The best way to combine scRNA-seq data with DAb-seq data would be to perform a CITE-seq experiment with an antibody panel that overlaps (at least in part) with the DAb-seq antibody panel. In this way, different single-cell RNA cell clusters can be linked to the DAb-seq compartments through sparse variant calls and through shared immunophenotype information. However, as we show in our manuscript, genotypic clones can manifest as distinct immunophenotypes, or a common immunophenotype can be shared by multiple genotypes. Therefore, the proposed method is not guaranteed to allow correct DNA-RNA assignments. The ideal solution would instead be to read out immunophenotype, genotype and transcripts from the same cell, a method that is yet to be developed.

To address the reviewer's point, we have amended the discussion. We thank the reviewer for voicing this comment as we believe the amended discussion emphasizes the power of our approach and the ways it can be extended to obtain ever deeper and more comprehensive single cell analysis datasets.

Reviewers' Comments:

Reviewer #1:

Remarks to the Author:

I think the authors have carefully and convincingly addressed my concerns. I think the manuscript is ready for publication

Reviewer #2:

Remarks to the Author:

The authors addressed all my comments thus I do not have further major concerns. However, during the revision time, two landmark studies have been published on the same topic:

- Nat Commun. 2020 Oct 21;11(1):5327, PMID: 33087716, "Clonal evolution of acute myeloid leukemia revealed by high-throughput single-cell genomics"

- Nature 2020 Nov;587(7834):477-482, PMID: 33116311, "Single-cell mutation analysis of clonal evolution in myeloid malignancies"

Since the results are very similar to the finding from this manuscript, the authors should discuss and cite these two articles.

Reviewer #3:

Remarks to the Author:

The authors have well addressed my comments.

Joint profiling of DNA and proteins in single cells to dissect genotype-phenotype associations in leukemia

Demaree and Delley, et al.

NCOMMS-20-36979

Revision #2

Point-by-point responses to reviewer comments are provided below.

Reviewer #1

Comment 1: I think the authors have carefully and convincingly addressed my concerns. I think the manuscript is ready for publication

Response: We are pleased that the reviewer has found the revised manuscript suitable for publication, and once again thank the reviewer for their time evaluating our work.

Reviewer #2

Comment 1: The authors addressed all my comments thus I do not have further major concerns. However, during the revision time, two landmark studies have been published on the same topic:
- Nat Commun. 2020 Oct 21;11(1):5327, PMID: 33087716, “Clonal evolution of acute myeloid leukemia revealed by high-throughput single-cell genomics”
- Nature 2020 Nov;587(7834):477-482, PMID: 33116311, “Single-cell mutation analysis of clonal evolution in myeloid malignancies”

Since the results are very similar to the findings from this manuscript, the authors should discuss and cite these two articles.

Response: We thank the reviewer for this helpful suggestion. These publications indeed demonstrate the power of the technology to glean insights into hematological disease from larger patient cohorts. We have added these citations to the Discussion section of the manuscript.

Reviewer #3

Comment 1: The authors have well addressed my comments.

Response: We are pleased that the reviewer has found the revised manuscript suitable for publication, and once again thank the reviewer for their time evaluating our work.